# Internal audit quality and accounting information comparability: Evidence from China

**Guochao Liu[1], Jingyu Wang[2]\*, Yanhan Sun[3], Jianluan Guo[1], Yufei Zhao[1]**

**1** School of Business, Central University of Finance and Economics, Beijing, China, **2** School of Accountancy, Beijing Wuzi University, Beijing, China, **3** School of Business and Administration, Shandong University of Finance and Economics, Jinan, China

\* edu605@126.com

**Data Availability Statement:** All relevant data are within the manuscript and its Supporting information files.

**Funding:** Shandong Province Natural Science Foundation Youth Branch (grant numbers:

## Abstract

This study employs the impact and mechanism of internal audit quality on the comparability of corporate accounting information. Using manually collected internal audit data, the study focuses on non-financial listed companies of A-share market spanning from 2007 to 2022 in China. The findings are as follows: (1) Internal audit significantly enhances the comparability of accounting information, the higher the quality of internal audit, the more pronounced its contribution. This conclusion remains robust after conducting endogeneity tests. (2) Mechanism testing reveals that internal audit enhances accounting information comparability through two channels: mitigating agency costs and improving financial information transparency. (3) Moderation effect test proves that the quality of internal control and the high-tech industry will strengthen the positive relationship between internal audit and comparability of accounting information, while the degree of industry competition and the level of capital occupation by major shareholders will weaken the positive relationship.

## 1. Introduction

The quality of accounting information in enterprises can alleviate the information asymmetry between internal and external investors, and plays a pivotal role in enhancing the efficiency of capital market resource allocation and safeguarding investor interests. Agency conflicts between owners and managers are important factors affecting the quality of accounting information [1, 2]. Previous studies have shown that internal control [3, 4], internal audit [5–7], and external audit [8, 9] are three effective mechanisms for reducing agency costs and improving the quality of accounting information. Despite the gradual improvement in the Chinese capital market system and legal environment, along with notable advancements in internal governance in enterprises, instances of accounting information violations persist due to inadequate internal oversight. For example, the illegal information disclosure by Changsheng Biotechnology (https://baijiahao.baidu.com/s?id=1619549428271560655&wfr=spider&for=pc), the financial fraud by Kangdexin (https://m.thepaper.cn/baijiahao_9396240), and the stock price manipulation by Guangzhou Anzhou (https://www.cs.com.cn/sylm/jsbd/201805/

ZR2020QG033,2020), Shandong Management College Research Sailing Program Fund (grant numbers: QH2021R01,2021), Beijing Municipal Commission of Education Social Science General Branch(grant numbers:SM202410037004,2024).

**Competing interests:** NO authors have competing interests.

t20180518_5806392.html?open_source=weibo_search) have all reduced the usefulness of information decision-making and are not conducive to investors making correct investment decisions.

While both external audit and internal control contribute to enhancing the quality of accounting information, internal audit possesses distinct advantages. Firstly, internal auditing, unlike external auditing, comprehensively assesses all management systems within a company and gathers audit evidence through informal methods. Additionally, internal audit staff are primarily internal employees of the enterprise, facilitating easier trust and support from audited personnel. Consequently, internal audit is more likely to exert governance effects and enhance information quality compared to external audit. Conversely, internal control faces challenges due to information asymmetry and increased transaction costs, hindering shareholders or ultimate controllers from obtaining effective information for direct management supervision. At this juncture, internal control lacks substantive independence. In contrast, internal audit, with its dual functions of confirmation and consultation, is better positioned to improve the quality of accounting information. Enhanced professional personnel, increased project activities, and effective feedback mechanisms within internal audit institutions are instrumental in bolstering their independence and authority for effective management supervision, thereby enhancing the quality of accounting information. Moreover, leveraging the expertise of audit committees to provide advisory functions further enhances the quality of accounting information. Hence, re-evaluating the internal audit function and improving accounting information quality are imperative.

Comparability is an important characteristic of accounting information quality, which can reduce the cost of debt financing for enterprises [10, 11] and reduce the risk of stock price collapse [12]. Compared to accounting information quality, such as prudence and timeliness, which only reflect the company's own characteristics, comparability can provide incremental information about the industry in which the company operates, reducing the cost of information collection for report users [13]. Firstly, comparability is a key indicator that reveals the unique differences between a company and its competitors in the same industry. It helps investors identify the unique risks of the company and can also assist accounting information users in directly comparing the financial status differences of companies in the same industry [14]. Secondly, comparability can present common characteristics of the industry in which the company operates, which helps clients to excavate and predict the accounting information of the target company by comparing the financial reports of other companies in the same industry [15]. Finally, comparability of accounting information is beneficial for companies in the same industry to enhance their competitiveness through mutual learning, thereby building a more stable and resilient supply chain [16]. Therefore, exploring how to improve the comparability of accounting information is of positive significance for enhancing the decision-making usefulness of external stakeholders [17] and promoting the optimal allocation efficiency of economic resources. Previous studies have found that auditors using unique audit methods and styles can affect accounting information comparability [18–23]. However, due to the availability of data, there has been no research on how internal audit affects the comparability of accounting information as an internal supervision mechanism. From the perspective of the function of internal audit, effective implementation of internal audit can timely detect and correct fraudulent behavior in financial statements [24], and constrain management's improper selection of accounting policies [25]. Therefore, promoting the internal audit function can provide support for enterprises to effectively improve the comparability of accounting information.

However, there is controversy over the impact of internal auditing on the quality of accounting information. Based on agency theory, some studies suggest that internal auditing

can alleviate the principal-agent problem, reduce the level of earnings management, and improve the quality of accounting information by monitoring the self-interest behavior of executives [5–7]. However, the executive rights theory suggests that the promotion and other incentives of the head of the internal audit department are constrained by the company's management, and the internal audit department may not be able to maintain independence and play a governance role in the self-interest behavior of the management, ultimately failing to improve the quality of accounting information [26]. Theoretical debates and practical needs require in-depth research based on Chinese listed companies.

There are two main reasons why we focus on the Chinese capital market. On the one hand, as the world's largest emerging market, China's legal and external market supervision mechanisms are not yet sound [27]. Studying the impact of internal auditing on the comparability of accounting information at this time can help us better understand the benefits of strengthening corporate internal governance supervision mechanisms in emerging capital markets. On the other hand, the accounting information quality of Chinese listed companies varies, and the effectiveness of the capital market needs to be improved. The improvement of comparability can improve the information environment of enterprises and reduce the degree of information asymmetry, thereby safeguarding the interests of stakeholders such as investors and creditors, and optimizing resource allocation. Therefore, studying the factors influencing comparability is of great significance for shaping a favorable information environment in emerging market countries.

Therefore, based on the data of Chinese A-share listed companies from 2007 to 2022, this article studies the impact of internal audit on the comparability of accounting information. Firstly, we found that internal audit can improve the comparability of accounting information. Secondly, we investigated the channels through which internal audit affects comparability and found that internal audit improves comparability by reducing agency costs and enhancing financial information transparency. Finally, the moderating effect test proves that the quality of internal control and the high-tech industry will strengthen the positive relationship between internal audit and comparability, while the degree of industry competition and the level of large shareholders' capital occupation will weaken the positive relationship.

Although Guo and Guo (2021) [28] have found that the mechanism by which internal control improves information comparability is also to reduce agency costs and improve information quality, we believe that this does not hinder the effectiveness of internal audit functions, as internal audit and internal control are interdependent and coupled [29]. On the one hand, high-quality internal auditing can improve the soundness and effectiveness of internal controls. On the other hand, sound internal controls can provide a solid foundation for internal auditing [30]. This is because a well-designed and operational internal control system can provide a good foundation for risk control and information credibility, providing a reliable working foundation for internal audit. It also puts forward direction and requirements in determining the scope and focus of auditing, selecting audit procedures, methods, and other aspects. Therefore, the higher the internal quality level, the more conducive it is to the performance of the internal audit function.

The research contribution includes two aspects: theoretical contribution and practical contribution. In terms of theoretical contribution, on the one hand, we have clarified the controversy surrounding the role of existing theories in internal auditing. The agency theory believes that internal audit is an important mechanism for improving accounting information supervision [5, 6], while the management rights theory suggests that the internal audit department may collude with management [26]. The above theoretical contradictions need to be clarified. This article is based on the perspective of comparability of accounting information, opening up the black box of the mechanism of internal audit on comparability of accounting

information. It is found that internal audit can improve comparability of accounting information by reducing agency costs and improving financial information transparency, providing empirical evidence for theoretical debates. Meanwhile, we verified the impact mechanism of internal audit on comparability from multiple perspectives such as internal control and industry competition, and provides evidence of the interaction between internal audit and internal control. On the other hand, our research enriches the relevant research on the factors affecting the comparability of accounting information at the micro enterprise level. Previous literature mainly studied the impact of external auditors using unique audit methods and styles [18–23] on comparability, but they overlooked the strengthening effect of micro level governance tools of internal audit companies on comparability. Compared to external auditing, internal auditing can examine all management systems of a company, and informal methods can be used to collect audit evidence, affecting the behavior of corporate executives and thus affecting the comparability of accounting information. This study can make up for this deficiency and deepen the understanding of the internal audit supervision function.

In terms of practical contribution, on the one hand, this article helps enterprises improve the construction of internal auditing. Internal audit plays an important role in alleviating agency issues and improving the quality of financial information in enterprises. Therefore, enterprises can further improve their internal audit system, such as strengthening the degree of management's emphasis on the design of internal audit systems, and enhancing the independence and authority of the internal audit department. On the other hand, this article helps to improve the level of internal governance in enterprises. Suppressing agency costs and improving financial information transparency are important paths to improve the comparability of accounting information. Therefore, enterprises should reasonably guide managers to reduce agency problems, which is the key enterprises' benign development. In addition, our research helps to improve the comparability of corporate accounting information, thereby safeguarding the interests of stakeholders such as investors and creditors, and improving the efficiency of investor decision-making.

The subsequent sections are organized as follows: the second segment comprises a comprehensive literature review and the formulation of hypotheses. The third segment delineates the research design. The fourth segment presents empirical results. The fifth segment undertakes further analysis and the final segment encapsulates the conclusion.

## 2. Literature review and hypothesis development

### 2.1 Research on economic consequences of internal audit

Internal audit is an important component of corporate governance, which can improve the quality of financial information [5–7, 31–33] by effectively monitoring the behavior of managers [24], and reduce the level of information asymmetry [34], thereby bringing a series of positive governance effects. Previous studies have found that internal audit can suppress earnings management, reduce internal control deficiencies [35] and violations [36, 37], lower risk levels [38], and improve profitability [39, 40]. However, some studies also suggest that the promotion and other incentives for the head of the internal audit department are constrained by the company's management. For the management's self-interest behavior, the internal audit department may not be able to maintain independence and play a governance role in its self-interest behavior, ultimately failing to improve the quality of accounting information [26].

In addition, studies have shown that internal audit can increase the dependence of external audit institutions on internal audit functions [41], reduce external audit costs [42–44], and improve the timeliness of external audits [45].

## 2.2 Research on the influencing factors of accounting comparability

Previous studies have shown that the comparability of accounting information is mainly affected by macro and enterprises' own factors. At the macro level, studies mainly found that the unified implementation of international accounting standards can significantly improve comparability [46–48]. Lin et al. (2019) [49] further compared the impact of Germany's adoption and convergence on comparability, and found that both methods improved comparability. However, the effect of the implementation of IAS in improving comparability depends on the strictness of the implementation requirements of IAS in various countries [50], so the convergence of standards does not necessarily lead to the comparability of accounting information [51]. In addition, Dhole et al. (2023) [52] found that economic policy uncertainty would diminish the comparability of accounting information.

In terms of enterprises' own factors, existing studies have found that the unique audit methods and styles adopted by auditors will affect the comparability of accounting information [18–23]. For example, Chen et al. (2020) [20] found that the more experienced the auditor, the higher comparability in the audited financial reports. At the same time, studies have also proved that strong CEOs [53] and newly appointed CEOs [54] can improve the comparability of accounting information, and Comparability increases with the presence and strength of joint ownership [55]. In addition, Dhole et al. (2015) [56] found that machine-readable information options (XBLR authorization) also affect comparability, and the study found that the comparability of accounting information declined in the first few years after the task.

## 2.3 Literature commentary

Through the review of existing research, it can be found that the researchers have achieved rich results in the study of factors affecting the comparability of accounting information and the economic consequences of internal auditing, providing important references for our research. However, it can be found that there is still controversy over whether internal auditing can improve the quality of accounting information. Meanwhile, in the context of increasingly strengthened audit supervision, few studies have examined the impact and mechanism of internal audit on the quality of accounting information from the perspective of comparability. We attempt to reveal the mechanism black box between internal audit and comparability from both theoretical logic and empirical analysis, providing policy inspiration for optimizing the path of internal audit construction, promoting the improvement of corporate governance level, and improving the quality of information disclosure further.

## 2.4 Theoretical hypothesis

**2.4.1 Internal audit level and comparability of accounting information.** Bath et al. (2012) [17] pointed out that external legal environment, accounting standards, and the interpretation and implementation of accounting standards by financial personnel can affect the quality of disclosed accounting information. According to agency theory, the current accounting standards give enterprises a certain degree of autonomy, and the management, as the direct responsible person, has a direct impact on the accounting selection, recognition, measurement, and other aspects of the enterprise. Therefore, whether there is self-interest in the selection of accounting policies and methods by management becomes the most direct factor affecting the comparability of accounting information. When management engages in opportunistic behavior, they use the given rights and space to choose accounting treatment policies and methods that are most conducive to completing transactions, and manipulate the reporting process of accounting information to reduce its comparability. Previous studies have shown that internal audit, as the main supervisory mechanism for corporate governance [36],

can to some extent prevent opportunistic behavior by management [24], alleviate agency conflicts [25], and improve financial information transparency [5–7]. Therefore, we believe that internal auditing can improve comparability by alleviating agency conflicts and improving information quality. Based on the above analysis, this article proposes the following hypothesis:

H1: Internal audit could improve comparability of accounting information. Specifically, higher levels of internal audit correlate with heightened accounting information comparability.

**2.4.2 The mechanism of agency problems.** According to the agency theory, internal auditing can suppress the self-interest and opportunistic behavior of executives, providing supervision and management support for improving the comparability of accounting information. The rapid development of the digital economy has led traditional enterprises to face a fierce market competition environment, which is easily affected by funding and liquidity risks, leading to a decline in profitability and generating strong earnings pressure [57]. In order to cope with this kind of earnings pressure, companies that engage in earnings manipulation may choose unreasonable accounting policies to increase accounting profits, resulting in significant differences in accounting procedures and methods used by management who have not engaged in earnings manipulation when accounting for the same economic transactions, thereby reducing comparability. The implementation of internal audit functions can effectively reduce management's earnings manipulation behavior in accounting policy selection [25]. Internal audit achieves effective supervision of managers through its supervisory function [36], which can reduce the first type of agency costs [58], suppress management's self-interest and opportunistic behavior [37], reduce management's discretion in accounting policy choices, and thereby improve comparability. Specifically, internal audit uses systematic and standardized methods to review and evaluate the appropriateness and effectiveness of the management's manipulation of financial statement numbers in collaboration with other functional personnel, increasing the likelihood of management's arbitrary changes in accounting policies being detected, deterring the self-interest and opportunistic behavior of executives, enhancing management's prudence in accounting estimates and policy choices, thereby ensuring consistency in accounting policy choices, and ultimately improving comparability. Based on the above analysis, this article proposes the following hypothesis:

H2: Internal audit could improve comparability of accounting information by mitigating agency costs.

**2.4.3 The mechanism of financial information transparency.** Internal auditing can suppress the level of earnings management in enterprises [25], improve the quality of financial information [5–7, 31–33], reduce the drawbacks caused by information asymmetry [34], and create a good information environment for improving the comparability of accounting information. Comparability is an important source of information for information users to make decisions [17], which can reduce the cost of information collection and processing, improve the usefulness of decision-making, and thus protect their legitimate interests. However, in practice, due to information asymmetry between information producers and users, information producers rely on their information advantages and discretion in implementing accounting standards to interfere with the production and disclosure of accounting information, thereby reducing comparability. As Zhou and Yang (2018) [59] found in their research, insiders with information advantages who hide internal transaction behavior will reduce comparability. De Franco (2011) [13] also found that the more transparent the information

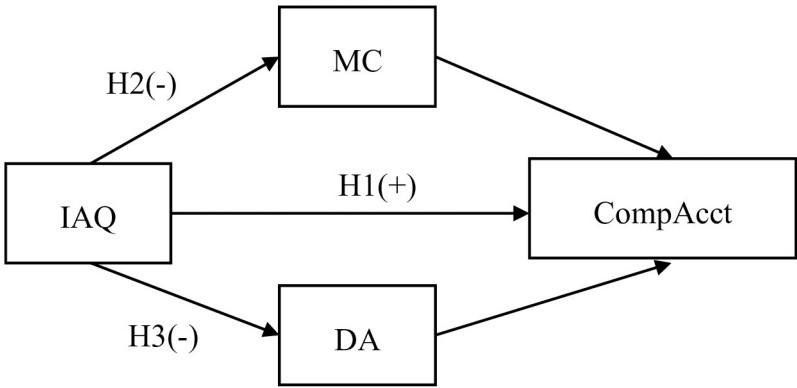

**Fig 1. Hypothetical model.** https://doi.org/10.1371/journal.pone.0310959.

environment of a company, the higher the comparability of accounting information. It can be seen that improving the information environment of enterprises helps to enhance comparability. Internal audit has a special comparative evaluation function, which can improve the internal information environment of enterprises, enhance the quality of financial information [5–7], correct erroneous information in financial statements, promote the release of true financial data, and ensure the production of high-quality comparable accounting information. At the same time, as the transparency of financial information increases, the likelihood of executives using complex earnings management to conceal bad news decreases. This invisibly weakens the inherent motivation of management to selectively disclose information, increases their prudence in accounting estimates and policy choices, and ensures consistency in accounting policy choices, thereby improving comparability. Based on the above analysis, this article proposes the following hypothesis:

H3: Internal audit could improve comparability of accounting information by enhancing financial information transparency (Fig 1).

## 3. Research design

### 3.1 Sample and data

Due to the international convergence of accounting standards in China in 2007, this study selected A-share listed companies in Shanghai and Shenzhen from 2007 to 2022 as the research sample. We manually organize internal audit level data from sources such as annual reports, internal control evaluation reports, and company internal audit systems. The other sample data is sourced from the CSMAR and WIND databases. On the basis of the initial sample (47698), the following samples were excluded: (1) The sample of financial listed companies (1416) was excluded because their asset structure and profit model are different from general enterprises; (2) Excluding the sample of listed companies with trading status ST, ST *, and PT (2747) due to the heterogeneity of their financial status; (3) Exclude a sample of 4426 listed companies that have undergone mergers and acquisitions or changes in their main business, as this would result in changes in performance; (4) Exclude a sample of 409 companies that have been listed for less than a year, as they may be affected by IPO effects; (5) Exclude a sample of 6961 listed companies that have issued both B-shares and H-shares simultaneously, as there may be differences in overseas legal regulation compared to domestic ones; (6) Exclude a

sample of 13510 listed companies with incomplete data. A total of 18229 valid samples (47698-1416-2747-4426-409-6961-13510) were obtained in the end. To eliminate the influence of extreme values, this study subjected all continuous variables to winsorize at the 1% and 99% levels.

## 3.2 Variables

**3.2.1 Explained variable.** This study delineates accounting systems as the conversion process for generating financial statements based on enterprise economic transactions, drawing upon De Franco et al. (2011) [13]. When companies i and j share similar accounting systems and engage in comparable economic transactions, their resulting financial statements exhibit similarity. Using data from company i over 16 consecutive quarters preceding the t period, this study employs accounting earnings (Earnings$_{i,t}$, calculated as quarterly net profit divided by the beginning equity market value) as the dependent variable and quarterly stock returns (Return$_{i,t}$) as the explanatory variable to estimate accounting system of company i during the t period. Recognizing the asymmetry in the treatment of good news and bad news (i.e., gains and losses represented by positive and negative stock returns), the enterprise accounting system is more timely in confirming bad news than good news. This article uses Basu's (1997) accounting robustness model [60] to modify the original model. In De Franco et al.'s (2011) model [13], the dummy variable of stock returns (Neg$i$, t) and the cross term of stock returns (Neg$_{i, t}$ × Returnn$_{i, t}$) are added to estimate the company's accounting system. The regression model is as follows:

$$Earnings_{i,t} = \alpha_i + \beta_i Return_{i,t} + c_i Neg_{i,t} + d_i Neg_{i,t} \times Return_{i,t} + \varepsilon_{it} \tag{1}$$

In Eq (1), Neg$_{i,t}$ is represented as 1 if the quarterly stock return is negative, and 0 otherwise. For estimating the comparability of accounting information, the assumption is made that two companies share identical economic transactions, denoted as Return$_{i,t}$. The expected earnings are then computed utilizing the accounting systems of both Company i and Company j.

$$E(Earnings)_{i,i,t} = \hat{\alpha}_i + \hat{\beta}_i Return_{i,t} + \hat{c}_i Neg_{i,t} + \hat{d}_i Neg_{i,t} \times Return_{i,t} + \varepsilon_{it} \tag{2}$$

$$E(Earnings)_{i,j,t} = \hat{\alpha}_j + \hat{\beta}_j Return_{i,t} + \hat{c}_j Neg_{i,t} + \hat{d}_j Neg_{i,t} \times Return_{i,t} + \varepsilon_{it} \tag{3}$$

Eq (2) represents the expected earnings calculated using the accounting system of company i during period t, whereas Eq (3) represents the expected earnings calculated based on the accounting system of company j during the same period. The comparability of accounting information between Company i and Company j (CompAcct$_{i,j,t}$) is defined as the inverse of the average absolute difference between the two companies' expected earnings.

$$CompAcct_{i,j,t} = -\frac{1}{16} \times \sum_{t-15}^{t} |E(Earnings)_{i,i,t} - E(Earnings)_{i,j,t}| \tag{4}$$

According to the above method, the comparability of accounting information between company i and other firms within the same industry is calculated. Subsequently, the calculated comparability value is sorted from largest to smallest, and the average CompAcct of all combinations is calculated as the dependent variable in the main regression of this paper. The larger the CompAcct, the stronger the comparability of accounting information.

**3.2.2 Interpreted variable.** Based on the studies of Ismael and Kamel (2021) [61], Wei and Yuan (2014) [62], IAQ quality is comprehensively measured. This comprehensive measurement (4 points) captures four different IAQ quality characteristics (1 point each), namely

the membership mode of the internal audit department (IAModel), the scope of responsibility of the internal audit department (IADuty), whether the chairman and general manager are under a single individual (Dual), and whether firms are audited by one of the Top 10 Audit Firms (Big10). The internal audit level IAQ is calculated by summing model (1). The larger the value of IAQ, the higher the internal audit level of the company. At the same time, the samples with missing data are eliminated.

$$IAQ_{i,t} = IAModel_{i,t} + IADuty_{i,t} + Dual_{i,t} + Big10_{i,t} \qquad (5)$$

Specifically, "IAModel" represents the organizational structure of the internal audit department, reflecting its operational mode. When the internal audit department operates under dual leadership, involving both the board of directors and the management, under the supervisory board, under the board of directors, or the audit committee, "IAModel" is assigned a value of 1. Conversely, if the internal audit department operates under the management or the finance department, "IAModel" is assigned a value of 0. This categorization is based on the premise that a higher affiliation of the internal audit department corresponds to an elevated organizational status, indicating increased independence, authority, and resources. Consequently, this facilitates the comprehensive execution of its functions [24].

"IADuty" delineates the scope of responsibilities assigned to the internal audit department, serving as an indicator of executive management's endorsement of the department. The value of "IADuty" is calculated by dividing the total number of special audits, financial audits, internal control evaluations, and consulting tasks undertaken by the internal audit department by three. This approach is based on the belief that the internal audit function should encompass more than just financial audits and should receive strong backing from senior management. A broader scope of the internal audit function indicates greater corporate support [63].

The variable "Dual" assesses whether the roles of the chairman of the board and the general manager are combined, reflecting the influence of the board of directors. When the positions of chairman and general manager are separate, "Dual" is assigned a value of 1; otherwise, it is assigned a value of 0. This assignment is based on the understanding that consolidating these two positions leads to an excessive concentration of power in the general manager, impeding the effective functioning of the board of directors. Consequently, this situation contributes to the partial failure of the internal audit function.

The variable "Big10" indicates whether the firm undergoes an audit by one of the Top 10 Audit Firms, serving as an indicator of the external audit's influence. A value of 1 is assigned if the firm engages a Top 10 Audit Firm for the audit, and 0 otherwise. This assignment is based on the recognition of a complementary relationship between internal and external audits [64]. Firms committed to robust corporate governance are more likely to adopt comprehensive internal auditing practices and are inclined to engage high-quality external audits [65].

**3.2.3 Control variables.** In terms of controlling variables, we draw on the research conducted by Ege et al. (2020) [19] and Peng et al. (2023) [55]. The fundamental-level variables include company size (Size), solvency (Lev), years of establishment (FirmAge), profitability (ROA), board of directors (Board), and cash flow (Cashflow). On the governance level, we control for the balance of equity (Balance) and concentration of equity (Herfindahl). At the market level, we include Tobin Q and price-to-earnings ratio (EPS) as control variables. Additionally, annual effects (YEAR) and industry effects (IND) are controlled.

The variable definition is shown in Table 1.

**Table 1. Variable definitions.**

| Variable | | Indicators |
|---|---|---|
| Explained variable | CompAcct | Comparability of accounting information, calculated as above |
| Interpreted variable | IAQ | Level of internal audit, calculated with reference to Ismael and Kamel (2021), Wei and Yuan (2014). |
| Control variables | Size | Firm size, calculated as ln (Total assets) |
| | Lev | The leverage ratio, calculated as Total liabilities / Total assets |
| | ROA | Return on assets, calculated as Net profit / Total assets |
| | Board | Board size, calculated as ln (Number of directors) |
| | Cashflow | Cash flow ratio, calculated as Net cash flows from operating activities / Total assets |
| | TobinQ | Tobin's Q, calculated as Market value / Replacement value |
| | Balance | Shareholding checks and balances, calculated as the sum of the shareholdings of the second through fifth largest shareholders over the shareholding of the first largest shareholder |
| | FirmAge | Years of establishment, calculated as ln (Current year—Establishment year +1) |
| | Herfindahl | Equity concentration, calculated as the sum of the squares of the shareholdings of the company's top 3 largest shareholders |
| | YEAR | Year dummy variables |
| | IND | Industry dummy variables |

## 3.3 Model specification

In order to examine the impact of internal audit on the comparability of accounting information, following the ideas of Peng et al. (2023) [55] and Francis et al. (2014) [18], we have established the following model based on the following reasons: firstly, the OLS model has fewer related constraints, which can effectively construct the correlation between the explanatory variable and the dependent variable. Secondly, by referring to previous literature, the main factors affecting the comparability of corporate accounting information were added to the model, effectively avoiding the interference of relevant factors and enhancing the robustness of the results. Thirdly, we controlled for annual fixed effects and industry fixed effects in the model to avoid the interference of features that do not change over time and industry on the research results. The specific model is as follows:

$$\text{CompAcct}_{i,t} = \alpha_0 + \alpha_1 \text{IAQ}_{i,t} + \sum \text{Control}_{i,t} + \alpha_2 \text{YEAR} + \alpha_3 \text{IND} + \varepsilon_{i,t} \tag{6}$$

In this model, CompAcct serves as the proxy variable for accounting information comparability, IAQ quantifies the internal audit level of listed companies, $\sum$ Control represents the set of control variables, and YEAR and IND denote year fixed effects and industry fixed effects, respectively. To mitigate the potential influence of heteroscedasticity and sequence-related issues in the Ordinary Least Squares (OLS) regression process on the conclusions, we employed robust P-values for the standard error of the regression coefficients in this study. Concurrently, cluster analysis at the industry level is used in the basic regression for the following reasons: First, the comparability of accounting information reflects the similarity or similarity of the same type of accounting items between different enterprises. Therefore, compared with the company level, the industry level is also an important factor affecting the distribution of random disturbance terms. Second, relevant literature research shows that factors at the industry level, such as the degree of industry competition, industry attributes and other endogenous factors have a direct impact on the comparability of accounting information.

**Table 2. Descriptive statistics.**

| Variable | N | Mean | SD | Min | Max |
|---|---|---|---|---|---|
| CompAcct | 18229 | -1.277 | 0.711 | -4.243 | -0.405 |
| IAQ | 18229 | 2.768 | 0.800 | 0.333 | 4 |
| Size | 18229 | 22.48 | 1.348 | 19.75 | 26.41 |
| Lev | 18229 | 0.473 | 0.199 | 0.0490 | 0.886 |
| ROA | 18229 | 0.0360 | 0.0620 | -0.210 | 0.223 |
| Board | 18229 | 2.159 | 0.202 | 1.609 | 2.708 |
| Cashflow | 18229 | 0.0490 | 0.0700 | -0.170 | 0.249 |
| TobinQ | 18229 | 1.993 | 1.275 | 0.844 | 8.197 |
| Balance | 18229 | 0.636 | 0.581 | 0.0240 | 2.679 |
| FirmAge | 18229 | 2.910 | 0.317 | 1.609 | 3.497 |
| Herfindahl | 18229 | 0.155 | 0.118 | 0.0120 | 0.562 |

## 4. Empirical results

### 4.1 Descriptive statistics

Table 2 presents the descriptive statistical results of the main variables, reflecting the following main descriptive statistical characteristics: firstly, the average value of the dependent variable (CompAcct) is (-1.28), indicating that there is still significant improvement space in the comparability of accounting information among the sample companies in terms of average significance. Meanwhile, CompAcct fluctuates between (-4.24) and (-0.41), indicating significant differences in the comparability of accounting information among the sample companies. Secondly, the explanatory variable IAQ fluctuates between 0.33 and 4, with a variance of 0.8, reflecting significant differences in the internal audit level of the sample companies. Thirdly, the descriptive statistical characteristics of the relevant control variables are basically consistent with the characteristics of the references. The above features provide an effective sample basis for further research.

### 4.2 The impact of internal audit on accounting comparability

Table 3 reports the regression results of the impact of internal audit on the comparability of accounting information. Column (1) shows the regression results without any control variables, and the coefficient of CompAcct is 0.054, showing a significant positive correlation at the 1% level. Column (2) shows the fixed effects regression results that only include industry and year. The CompAcct coefficient is 0.035, which has significant statistical significance at the 1% level. Column (3) displays the regression results for all control variables. The CompAcct coefficient is 0.020, showing a significant positive correlation at the 5% level. The results indicate that internal auditing can improve comparability. This result preliminarily confirms hypothesis H1.

Furthermore, this article attempts to conduct in-depth research based on the four dimensions of constructing IAQ, in order to identify which factors are the key driving factors affecting the comparability of accounting information. The specific results are listed in columns (4) —(7), and the results in column (4) indicate that the coefficient of IAModel is significantly positive at the 1% level, indicating that the membership mode of the internal audit department is a key factor affecting comparability in enterprises. Similarly, the coefficient of variable IAduty is also significantly positive at the 1% level, indicating that the larger the scope of internal audit responsibilities, the more conducive it is to improving comparability. However, in

**Table 3. The impact of internal audit on accounting comparability.**

| Variable | (1) | (2) | (3) | (4) | (5) | (6) | (7) |
|---|---|---|---|---|---|---|---|
| | CompAcct | CompAcct | CompAcct | CompAcct | CompAcct | CompAcct | CompAcct |
| IAQ | 0.054*** | 0.035*** | 0.019** | | | | |
| | (0.00) | (0.00) | (0.02) | | | | |
| IAModel | | | | 0.048*** | | | |
| | | | | (0.00) | | | |
| IADuty | | | | | 0.053*** | | |
| | | | | | (0.00) | | |
| Dual | | | | | | -0.012 | |
| | | | | | | (0.26) | |
| Big10 | | | | | | | 0.000 |
| | | | | | | | (0.11) |
| Size | | -0.134*** | -0.103*** | -0.101*** | -0.101*** | -0.104*** | -0.104*** |
| | | (0.00) | (0.00) | (0.00) | (0.00) | (0.00) | (0.00) |
| Lev | | -0.455*** | -0.436*** | -0.437*** | -0.438*** | -0.433*** | -0.434*** |
| | | (0.00) | (0.00) | (0.00) | (0.00) | (0.00) | (0.00) |
| ROA | | 2.905*** | 2.417*** | 2.411*** | 2.409*** | 2.427*** | 2.427*** |
| | | (0.00) | (0.00) | (0.00) | (0.00) | (0.00) | (0.00) |
| Board | | 0.131** | 0.069 | 0.075*** | 0.075*** | 0.073*** | 0.070*** |
| | | (0.01) | (0.11) | (0.00) | (0.00) | (0.00) | (0.00) |
| Cashflow | | -0.949*** | -0.703*** | -0.701*** | -0.701*** | -0.697*** | -0.697*** |
| | | (0.00) | (0.00) | (0.00) | (0.00) | (0.00) | (0.00) |
| TobinQ | | 0.009 | 0.006 | 0.005 | 0.005 | 0.005 | 0.005 |
| | | (0.39) | (0.57) | (0.17) | (0.18) | (0.17) | (0.16) |
| Balance | | -0.072*** | -0.066*** | -0.065*** | -0.066*** | -0.064*** | -0.064*** |
| | | (0.00) | (0.00) | (0.00) | (0.00) | (0.00) | (0.00) |
| FirmAge | | -0.366*** | -0.103 | -0.100*** | -0.100*** | -0.104*** | -0.105*** |
| | | (0.00) | (0.11) | (0.00) | (0.00) | (0.00) | (0.00) |
| Herfindahl | | -0.240*** | -0.214** | -0.204*** | -0.204*** | -0.210*** | -0.214*** |
| | | (0.00) | (0.03) | (0.00) | (0.00) | (0.00) | (0.00) |
| Constant | -1.427*** | 2.645*** | 1.359*** | 1.325*** | 1.313*** | 1.432*** | 1.434*** |
| | (0.00) | (0.00) | (0.00) | (0.00) | (0.00) | (0.00) | (0.00) |
| Industry FE | No | No | Yes | Yes | Yes | Yes | Yes |
| Year FE | No | No | Yes | Yes | Yes | Yes | Yes |
| Observations | 18,229 | 18,229 | 18,229 | 18229 | 18229 | 18229 | 18229 |
| adj. $R^2$ | 0.004 | 0.232 | 0.412 | 0.412 | 0.412 | 0.411 | 0.410 |

Note
*** $p < 0.01$
** $p < 0.05$
* $p < 0.1$. (P-value in parentheses)

contrast, the results in columns (6)—(7) indicate that dual roles and firm selection are not key factors affecting the quality of internal auditing in listed companies.

## 4.3 Robustness tests

**4.3.1 Changing the independent variable.**  To refine the measurement of accounting information comparability, the average value of the first four values is taken as the

**Table 4. Robustness tests.**

| Variable | (1) | (2) | (3) | (4) | (5) |
|---|---|---|---|---|---|
| | CompAcct4 | CompAcct | CompAcct | CompAcct | CompAcct |
| IAQ | 0.000* | | 0.020** | | 0.019** |
| | (0.05) | | (0.03) | | (0.03) |
| IAQ1 | | 0.038** | | | |
| | | (0.01) | | | |
| Commission | | | | 0.008** | |
| | | | | (0.03) | |
| Constant | 0.008*** | 1.374*** | 1.359*** | 1.301*** | 1.358*** |
| | (0.00) | (0.00) | (0.00) | (0.00) | (0.00) |
| Controls | Yes | Yes | Yes | Yes | Yes |
| Industry FE | Yes | Yes | Yes | Yes | Yes |
| Year FE | Yes | Yes | Yes | Yes | Yes |
| Observations | 18,229 | 18229 | 18,229 | 8160 | 18229 |
| adj. R2 | 0.223 | 0.411 | 0.411 | 0.464 | 0.411 |

Note

*** $p < 0.01$

** $p < 0.05$

* $p < 0.1$. (P-value in parentheses)

comparability measurement value of accounting information of company i and recorded as CompAcct4. The larger the value, the stronger the comparability of accounting information. The regression results in Column (1) of Table 4 demonstrate that the IAQ coefficient remains significantly positive, confirming the robustness of the benchmark regression results.

**4.3.2 Changing the dependent variable.** Reclassifying the internal audit level based on the annual industry median, we define the variable IAQ1. Assigning a value of 1 to samples with IAQ1 greater than or equal to the median, and 0 otherwise. Substituting IAQ with IAQ1 in model (6), the regression results in Column (2) of Table 4 indicate that the IAQ1 coefficient remains significantly positive, and the benchmark regression results exhibit robustness.

**4.3.3 Using industry and year bidirectional clustering standard errors.** Standard errors in bidirectional clustering can provide information about the accuracy and reliability of the clustering. Therefore, in order to obtain more robust regression results, we use the industry and year bidirectional clustering standard error pair model (6) for regression. The regression results of column (3) in Table 4 show that the coefficient of IAQ is still significantly positive, and the baseline regression results are still robust.

**4.3.4 Changing the independent variable.** In order to avoid the interference of internal audit level measurement bias on research results, explanatory variables are replaced. Specifically, the number of annual meetings of the audit committee of listed companies (Commission) is used as a proxy variable for the quality of internal audit of listed companies. The larger the indicator, the higher the quality of internal audit of listed companies. The specific results are shown in column (4) of Table 4, where Commission is significantly positive at the 5% level, indicating that the more audit committee meetings there are, the stronger the comparability of accounting information of listed companies, indicating that the quality of internal audit can improve comparability.

**4.3.5 Cluster analysis based on firm level.** In the basic regression, we adopted industry-level cluster analysis. In order to avoid the interference of company-level factors on the

research results, we further adopted company-level cluster analysis to conduct robustness test. The specific results are shown in column (5) of Table 4. The results show that the coefficient of variable IAQ is significantly positive at the 5% level, indicating that the correlation between internal audit and accounting information comparability remains notably positive on the basis of the company-level cluster analysis, thus further verifying the robustness of the basic regression results.

## 4.4 Addressing endogeneity

**4.4.1.PSM.** The study conducted in this article confronts potential endogeneity issues arising from data bias and confounding factors. To mitigate these concerns, the Propensity Score Matching (PSM) method is employed. Initially, Internal Audit Quality (IAQ) is categorized into high and low groups based on the annual industry median, creating a corresponding dummy variable, IAQ1. A value of 1 is assigned to IAQ1 for observations above the median, and 0 otherwise. Logistic regression is then applied to IAQ1, incorporating all control variables to derive scores for each observation. Using these scores, a radius matching method with a caliper of 0.05 is employed to match samples with high and low internal audit levels, generating matched samples. Subsequently, regression analysis is conducted on model (6). The regression results of the paired samples are reported in column (1) of Table 5. The coefficient of IAQ is still significantly positive, and the benchmark regression results are still robust.

**4.4.2 2SLS.** In order to alleviate the possible endogenous problem of mutual causation between comparability and internal audit, specifically, companies with higher accounting information comparability are more motivated to improve the level of internal audit, so this paper adopts the two-stage least square method (2SLS) to deal with it. In the first stage, the annual industry average of IAQ, excluding the target company, serves as the instrumental variable (IV). IV is regressed against the endogenous variable, yielding the predicted value, P_IAQ, detailed in column (2) of Table 5. In the second stage, P_IAQ is utilized as the independent variable to regress model (6), and the results are presented in column (3) of Table 5.

**Table 5. Endogeneity tests.**

| Variable | (1) | (2) | (3) |
|---|---|---|---|
| | CompAcct (PSM) | IAQ (IV-first stage) | CompAcct (IV-second stage) |
| IV | | 0.909*** | |
| | | (0.00) | |
| IAQ | 0.020** | | |
| | (0.03) | | |
| IMR | | | |
| P_IAQ | | | 0.543* |
| | | | (0.08) |
| Controls | Yes | Yes | Yes |
| Industry FE | Yes | Yes | Yes |
| Year FE | Yes | Yes | Yes |
| Observations | 18,225 | 18,229 | 18,229 |
| adj. $R^2$ | 0.411 | 0.045 | -0.322 |

Note

*** p<0.01

** p<0.05

* p<0.1. (P-value in parentheses)

Remarkably, the coefficient of P_IAQ remains significantly positive, reinforcing the support for H1 evident in the benchmark regression results.

## 4.5 Mechanism test

**4.5.1 Agency costs.** Internal audit plays a crucial role in reducing agency costs [58], curbing management's latitude in accounting policy choices, and ultimately enhancing comparability. Therefore, according to the study of Wen et al. (2004) [66], we conducted an intermediary effect analysis to test whether agency cost is the channel through which internal audit affects comparability.

We examine the impact of internal audit on comparability, as shown in model (6). Secondly, we build a model (7) to test the impact of internal audit on agency costs:

$$AC_{i,t} = \alpha_4 + \alpha_5 IAQ_{i,t} + \sum Control_{i,t} + \alpha_6 YEAR + \alpha_7 IND + \varepsilon_{i,t} \tag{7}$$

Lastly, using Model (8), we examined the impact of internal audit and first-type agency costs on the comparability of accounting information.

$$CompAcct_{i,t} = \alpha_8 + \alpha_9 IAQ_{i,t} + \alpha_{10} AC_{i,t} + \sum Control_{i,t} + \alpha_{11} YEAR + \alpha_{12} IND + \varepsilon_{i,t} \tag{8}$$

We measure agency costs using management fee rate ($MC_{i,t}$) [67]. The management expense rate can reflect the discretionary expenditure of managers. The higher management expense rate, the higher the agency cost.

The regression results, presented in Table 6, show the impact of the management expense rate as an intermediary variable. In Column (1), the IAQ coefficient of -0.001 is significantly negative at the 1% confidence level, signifying that IAQ diminishes the management expense rate (MC). In Column (2), the negative influence of MC on CompAcct is observed, while the IAQ coefficient is significantly positive at the 5% confidence level, indicating its potential to

**Table 6. Channel analysis.**

| Variable | (1) | (2) | (3) | (4) |
|---|---|---|---|---|
|  | MC | CompAcct | DA | CompAcct |
| IAQ | -0.001** | 0.018** | -0.001* | 0.018** |
|  | (0.04) | (0.04) | (0.10) | (0.05) |
| MC |  | -0.591*** |  |  |
|  |  | (0.01) |  |  |
| DA |  |  |  | -0.589*** |
|  |  |  |  | (0.00) |
| Constant | 0.277*** | 1.487*** | -0.049*** | 1.312*** |
|  | (0.00) | (0.00) | (0.00) | (0.00) |
| Controls | Yes | Yes | Yes | Yes |
| Industry FE | Yes | Yes | Yes | Yes |
| Year FE | Yes | Yes | Yes | Yes |
| Observations | 17,033 | 17,033 | 16,924 | 16,924 |
| adj. R² | 0.345 | 0.392 | 0.682 | 0.392 |

Note

\*\*\* p<0.01

\*\* p<0.05

\* p<0.1. (P-value in parentheses)

foster the above effects. These findings highlight a significant positive correlation between the management expense rate and the comparability of accounting information, suggesting that internal audit enhances comparability by curbing the management expense rate. According to the conditions for the formation of intermediary effect, columns (1) and (2) show that agency cost plays a partial intermediary role in the impact of internal audit on comparability. The above analysis supports hypothesis H2.

**4.5.2 Transparency of financial information.** Internal audit has been identified as a catalyst for enhancing financial transparency [5–7], reducing levels of earnings management [25], and ultimately elevating the comparability of accounting information. In line with the methodology proposed by Wen et al. (2004) [66], this study undertakes a mediation effect analysis to scrutinize whether financial information transparency serves as a channel through which internal audit influences the comparability of accounting information. The examination of internal audit's impact on accounting comparability is detailed in model (6). Subsequently, we introduce model (9) to assess the influence of internal audit on financial information transparency.

Internal audit has been recognized as a key factor in enhancing financial transparency [5–7], reducing levels of earnings management [25], and ultimately improving the comparability of accounting information. Following the methodology proposed by Wen et al. (2004) [66], this study employs a mediation effect analysis to investigate whether financial information transparency serves as a pathway through which internal audit affects the comparability of accounting information. The examination of internal audit's impact on accounting comparability is detailed in model (6). Subsequently, we introduce model (9) to evaluate the influence of internal audit on financial information transparency.

$$IT_{i,t} = \alpha_4 + \alpha_5 IAQ_{i,t} + \sum Control_{i,t} + \alpha_6 YEAR + \alpha_7 IND + \varepsilon_{i,t} \qquad (9)$$

Finally, Model (10) is employed to examine the combined effect of internal audit and financial information transparency on comparability.

$$CompAcct_{i,t} = \alpha_8 + \alpha_9 IAQ_{i,t} + \alpha_{10} IT_{i,t} + \sum Control_{i,t} + \alpha_{11} YEAR + \alpha_{12} IND + \varepsilon_{i,t} \quad (10)$$

We employ accrual earnings management (DA) [68, 69] as a metric for assessing financial information transparency. In general, a decrease in earnings management corresponds to an increase in the transparency of financial information. The regression outcomes are presented in Table 6: In Column (3), the IAQ coefficient of 0.001 is significantly negative, signifying that IAQ effectively diminishes the level of earnings management. In Column (4), DA is observed to have a negative impact on CompAcct, yet the IAQ coefficient is notably positive at the 5% level, endorsing the aforementioned effects. These findings highlight a substantial positive correlation between financial information transparency and the comparability of accounting information, illustrating how internal audit enhances accounting information comparability by fostering information transparency. Based on the criteria for establishing an intermediary effect, Columns (3) and (4) suggest that financial information transparency partially mediates the influence of internal audit on the comparability of accounting information. This analysis lends support to hypothesis H3.

We use accrual earnings management (DA) [68, 69] as a measure to evaluate financial information transparency. Generally, a reduction in earnings management corresponds to an improvement in the transparency of financial information. The regression results are presented in Table 6: In Column (3), the IAQ coefficient of 0.001 is significantly negative, indicating that IAQ effectively reduces the level of earnings management. In Column (4), DA is found to have a negative impact on CompAcct, while the IAQ coefficient is notably positive at

the 5% level, supporting the aforementioned effects. These results underscore a significant positive association between financial information transparency and the comparability of accounting information, demonstrating how internal audit enhances accounting information comparability by promoting information transparency. Based on the criteria for establishing a mediation effect, Columns (3) and (4) suggest that financial information transparency partially mediates the influence of internal audit on the comparability of accounting information. This analysis provides evidence in support of hypothesis H3.

## 5. Further analysis

### 5.1 Internal audit quality, internal control level and accounting comparability

Previous studies have shown that robust internal controls, by closely monitoring managerial actions, can reduce earnings management in a company [70] and improve the quality of accounting information disclosure [1, 71]. Therefore, the mechanism for enhancing accounting information comparability through internal control closely mirrors that of internal audit. Due to the interdependence between internal audit and internal control, as they intersect and interact [29], higher internal control quality enhances the effectiveness of internal audit functions, thereby facilitating the improvement of accounting information comparability.

To investigate the moderating effect of internal control, we utilized the DIB internal control rating index, denoted as InterCon, which reflects higher internal control quality. Ratings from AAA to D were assigned values from 8 to 1, respectively, and the interaction term InterCon×IAQ was included in model (6) for regression analysis. The results in column (1) of Table 7 show a significantly positive coefficient for the interaction term, indicating that internal control quality reinforces the positive relationship between internal audit and accounting information comparability.

**Table 7. Moderating effect analysis.**

| Variable | (1) | (2) | (3) | (4) |
|---|---|---|---|---|
| CompAcct | M = InterCon | M = CR4 | M = Occupy | M = Tech |
| IAQ | 0.011* | 0.021** | 0.027*** | 0.001 |
| | (0.09) | (0.02) | (0.00) | (0.87) |
| M | -0.028* | 0.049 | 0.168 | 0.042** |
| | (0.10) | (0.56) | (0.48) | (0.03) |
| M×IAQ | 0.023*** | 0.298*** | -0.600*** | 0.046*** |
| | (0.00) | (0.00) | (0.00) | (0.00) |
| Constant | 1.270*** | 1.345*** | 1.293*** | 1.284*** |
| | (0.00) | (0.00) | (0.00) | (0.00) |
| Controls | Yes | Yes | Yes | Yes |
| Industry FE | Yes | Yes | Yes | Yes |
| Year FE | Yes | Yes | Yes | Yes |
| Observations | 18,229 | 18,190 | 17143 | 17152 |
| adj. $R^2$ | 0.413 | 0.411 | 0.384 | 0.391 |

Note

*** $p < 0.01$

** $p < 0.05$

* $p < 0.1$. (P-value in parentheses)

## 5.2 Internal audit quality, industry competition and accounting comparability

The competitive product market, serving as a governance mechanism, exercises supervisory and constraining functions by conveying operational pressures to management. Firstly, heightened competition in the product market diminishes information asymmetry risks, alleviates principal-agent problems, and enhances information disclosure quality in the industry. External investors are more likely to uncover self-interested managerial behavior, leading to a corresponding reduction in management actions detrimental to external shareholders. Conversely, in a fiercely competitive product market, the presence of benchmark competition prompts company shareholders to scrutinize and supervise management decisions more closely. Managers jeopardizing shareholder interests may risk dismissal, motivating them to preserve their career prospects by curbing actions detrimental to the company's value [72, 73]. Consequently, heightened competition in the product market weakens enterprise agency problems, leading to increased financial information transparency.

Hence, compared to enterprises in less competitive industries, those in highly competitive sectors exhibit greater financial information transparency, reduced information asymmetry, and weaker agency issues. Essentially, heightened industry competition serves as an external oversight mechanism, bolstering enterprise value through enhanced financial information transparency and decreased agency problems. Since these pathways mirror the means by which internal audit elevates enterprise value, the scope for internal audit to fulfill its role is limited in the presence of robust industry competition. Consequently, the role of internal audit in enhancing enterprise value diminishes under high industry competition, whereas it strengthens in less competitive sectors. To examine the moderating impact of industry competition, we gauge industry concentration using CR4, calculated as the ratio of a company's main business income to that of the entire industry. A smaller index denotes higher industry competition. We introduced the cross-multiplication term CR4×IAQ into model (6) for regression analysis. The results in column (2) of Table 7 indicate a significantly negative coefficient for the cross-multiplication term, suggesting that industry competition weakens the positive relationship between internal audit and comparability.

## 5.3 Internal audit quality, capital appropriation by major shareholders and accounting comparability

As an emerging economic country, China's corporate equity is relatively concentrated, with the conflict of interests between major shareholders and small and medium-sized shareholders emerging as a predominant agency issue [74, 75]. The self-interest behavior of major shareholders can negatively impact economic transparency [76, 77], thereby potentially influencing the comparability of accounting information. The severity of the agency problem between major shareholders and small and medium-sized shareholders determines the extent of major shareholders' encroachment on the interests of smaller shareholders. Consequently, major shareholders are more likely to exploit their power to undermine the quality of internal auditing and restrict its role in enhancing information transparency.

To examine the influence of major shareholders on the relationship between internal audit and the comparability of accounting information, we utilize the capital occupancy rate of major shareholders, denoted as Occupy [78], to measure the agency cost between major shareholders and small and medium-sized shareholders. This is calculated using the ratio of other receivables to year-end total assets. A higher capital occupancy rate of major shareholders indicates greater agency costs. Examining the results of the interaction term (Occupy×IAQ), we observe a significantly negative coefficient. This suggests that in an environment characterized by high agency

problems among major shareholders, the positive impact of internal audit level on comparability is diminished. Comparing these results with previous findings, it appears that while internal audit helps mitigate agency problems between management and owners, it does not address the agency issues between major shareholders and small and medium-sized shareholders. Furthermore, this demonstrates the significant control exerted by major shareholders in China.

## 5.4 Internal audit quality, technological attributes and accounting comparability

Differences in information quality and disclosure levels exist among enterprises with varying technological attributes, potentially resulting in variations in the impact of internal audit levels on accounting information comparability. In comparison to non-high-tech enterprises, high-tech enterprises often experience more pronounced information asymmetry with the market [79]. Furthermore, high-tech enterprises tend to encapsulate substantial intangible resources such as core technology or knowledge, which are not fully disclosed within traditional accounting systems, potentially leading to lower information quality and comparability. This scenario may render internal audit functions more effective in enhancing information transparency and, consequently, improving comparability. As a result, this study anticipates a stronger positive correlation between internal audit levels and comparability in high-tech enterprises.

Thus, this study utilizes the Catalogue of Strategic Emerging Industries, the Classification of Strategic Emerging Industries (2012), and relevant documents from the Organization for Economic Cooperation and Development (OECD). The high-tech industries specified by the OECD include the following five industries: computer related industries, electronics industry, information technology industry, biopharmaceutical industry, and communication industry. In comparison with the Industry Classification Guidelines for Listed Companies (2012), the industry codes for high-tech listed companies are established. The three categories are manufacturing industry (C), information transmission, software and information technology services industry (I), and scientific research and technology services industry (M). The 19 categories include C25, C26, C27, C28, C29, C31, and C31. 2. C34, C35, C36, C37, C38, C39, C40, C41, I63, 64, I65, and M73. An interaction term (Tech×IAQ) is introduced for regression analysis. The research findings reveal that the coefficient of the interaction term is significantly positive at the 1% level, indicating that in high-tech enterprises, internal audit has a more pronounced effect on enhancing comparability.

## 6. Conclusion

The study employs the impact of internal audit on accounting information comparability, revealing that internal audit can significantly improve the comparability of accounting information. Analyzing the channels, it is evident that internal audit enhances accounting information comparability through the reduction of agency costs and the augmentation of financial transparency. The moderation effect test proves that the quality of internal control and the high-tech industry will strengthen the positive relationship between internal audit and comparability of accounting information, while the degree of industry competition and the level of capital occupation by major shareholders will weaken the positive relationship.

Our research has the following policy implications. First of all, internal audit plays an important role in alleviating corporate agency problems and improving the quality of corporate financial information. Therefore, enterprises should pay more attention to the construction of internal audit organization management system, innovate internal audit operation mechanism and practice environment, and ensure the efficient play of internal audit function. Second, the management of agency costs and the improvement of the quality of financial

information are still the key to the healthy development of enterprises. Therefore, enterprises should reasonably guide managers to reduce agency problems and take various measures to improve comparability. Finally, the economic consequences of internal audits are important. Therefore, when optimizing the level of internal governance, enterprises should make reasonable plans for the construction and optimization of internal audit system. This contributes to an effective monitoring mechanism and ultimately to sustainable development.

Nevertheless, our study has several limitations. Firstly, despite employing various methods to mitigate endogeneity, we acknowledge that this issue may not be entirely resolved. For instance, the measurement of internal audit quality may not be exhaustive. Secondly, our findings are derived from data sourced solely from China, and caution should be exercised when generalizing them to other countries due to differing institutional contexts. Despite these limitations, our research underscores the positive impact of internal audit levels on enhancing comparability, offering valuable insights for a deeper understanding of the governance role of internal audit and for enhancing the quality of accounting information in listed companies.

## Supporting information

**S1 Data.**
(ZIP)

## Author Contributions

**Conceptualization:** Guochao Liu, Jianluan Guo.

**Data curation:** Guochao Liu.

**Funding acquisition:** Guochao Liu, Jingyu Wang.

**Methodology:** Jingyu Wang.

**Software:** Jianluan Guo.

**Writing – original draft:** Guochao Liu, Yanhan Sun.

**Writing – review & editing:** Jingyu Wang, Yufei Zhao.

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
