## [Decision Letter · Decision Letter 0]

17 May 2024

PONE-D-24-12628Internal Audit Quality and Accounting Information Comparability: Evidence from ChinaPLOS ONE

Dear Dr. liu,

Thank you for submitting your manuscript to PLOS ONE. After careful consideration, we feel that it has merit but does not fully meet PLOS ONE’s publication criteria as it currently stands. Therefore, we invite you to submit a revised version of the manuscript that addresses the points raised during the review process.

We look forward to receiving your revised manuscript.

Kind regards,

Muhammad Kaleem Khan

Academic Editor

PLOS ONE

“Funding: Shandong Province Natural Science Foundation Youth Branch (grant numbers: ZR2020QG033,2020), Shandong Management College Research Sailing Program Fund (grant numbers: QH2021R01,2021).”

4. PLOS requires an ORCID iD for the corresponding author in Editorial Manager on papers submitted after December 6th, 2016. Please ensure that you have an ORCID iD and that it is validated in Editorial Manager. To do this, go to ‘Update my Information’ (in the upper left-hand corner of the main menu), and click on the Fetch/Validate link next to the ORCID field. This will take you to the ORCID site and allow you to create a new iD or authenticate a pre-existing iD in Editorial Manager. Please see the following video for instructions on linking an ORCID iD to your Editorial Manager account: https://www.youtube.com/watch?v=_xcclfuvtxQ.

Additional Editor Comments:

Dear Authors

Thank you very much for submitting the manuscript. You can see one reviewer 3 recommended rejection. Still I invite you to address the concerns raised by reviewer 2 and 1. I expect you will make satisfactory improvements in the revised versions to enable it for publication. While revising, also address the concerns raised by reviewer 3.

Regards

Dr Kaleem

Reviewers' comments:

Reviewer's Responses to Questions

**Comments to the Author**

1. Is the manuscript technically sound, and do the data support the conclusions?

Reviewer #1: Yes

Reviewer #2: Yes

Reviewer #3: Yes

2. Has the statistical analysis been performed appropriately and rigorously? 

Reviewer #1: Yes

Reviewer #2: No

Reviewer #3: Yes

3. Have the authors made all data underlying the findings in their manuscript fully available?

Reviewer #1: Yes

Reviewer #2: Yes

Reviewer #3: Yes

4. Is the manuscript presented in an intelligible fashion and written in standard English?

Reviewer #1: Yes

Reviewer #2: Yes

Reviewer #3: No

5. Review Comments to the Author

Reviewer #1: The article "Internal Audit Quality and Accounting Information Comparability: Evidence from China", is a pretty good study, but some of the following issues need to be clarified:

- Clarify and explain more deeply the theories related to the model and variables in the research model.

- Considering Internal Audit Quality, dividing into two groups with good quality and no quality, the comparison will be clearer.

Reviewer #2: Referee report for “Internal Audit Quality and Accounting Information Comparability: Evidence from China”

(Manuscript number: PONE-D-24-12628)

Summary

This paper examines the impact of internal audit quality on corporate accounting information comparability. By using Chinese A-share firms in 2007-2022 as the sample, the author(s) show that internal audit enhances information comparability by mitigating agency costs and improving financial information transparency. Such effect is stronger in firms with more capital cooperation by the controlling shareholder and high-tech companies, and when internal control quality is higher. The effect is weaker when firms are in industries with fiercer competition. Overall, this paper studies an important question about the relationship between internal audit and accounting information comparability. However, theoretical hypotheses and empirical design might need substantial development. My detailed comments are as follows. I hope authors find them useful.

Comments

1. This paper investigates the influence of internal audit quality on accounting information comparability. As indicated by the literature review, rich studies have examined how internal audit quality affects accounting information quality in terms of earnings management (Ege et al., 2022). It would be necessary to highlight the value and importance of information comparability and how it differentiates from other dimensions of information quality.

Meanwhile, although Endrawes et al. (2020) study audit committee characteristics, it is closely related to internal audit quality which relies on the function of the audit committee. Other studies (e.g., Guo and Guo, 2021; Li et al., 2022) examine the relationship between internal control quality and information comparability. In particular, Guo and Guo (2021) contend that the mechanisms of how internal control increases information comparability are reduced agency costs and improved information quality, which is the same as this paper.

Given the essential links between internal control and internal audit, authors might also want to explicitly illustrate how this paper extends the discussions in the literature, which is highly desirable for the paper to sharpen the contribution.

2. Authors suggest that internal audit can increase monitoring on management, which then improves information comparability. However, given the concentrated ownership structure in China, it is likely that the internal audit functions in different channels as those in markets with dispersed ownership structures. Authors can incorporate the unique features of Chinese firms in the theoretical analyses.

3. There are some other concerns with the empirical execution.

(1) Why does the sample period start from 2007? Why not cluster standard errors at the firm level in the baseline model? Authors may want to explain the sample selection.

(2) The authors used a consolidated measure of internal audit quality based on the affiliation model, the responsibility scope, the dual position of chairman and CEO, and whether the firm is audited by a Top 10 Audit Firm. As those dimensions are not similarly constructed by using dummy variables, authors can examine the impact of each dimension separately. This can also help explore the heterogeneous effects among them.

(3) To ensure the robustness of the results, authors can also use other measures of internal audit quality. For example, the factors include the effectiveness of the audit committee proxied by the number of audit meetings, committee size, and background of committee members (e.g., financial and accounting expertise).

(4) Authors shall recheck the data and results. It seems odd that the coefficient is significant at 10% when t=1.69 in Table 6 while it is insignificant when t=1.71 in Table 3. Given the confusing notations, it is hard to make conclusions from the results.

(5) For the subsample analyses, authors shall report the results of the between-group difference tests.

(6) Are instruments the same for the Heckman and the 2SLS tests? It seems the target company is excluded in calculating the average mean in 2SLS while it is not when conducting the Heckman two-stage estimation. Authors may want to elaborate on the calculation of instruments.

4. There are errors in writing. For example, the authors state that “internal control quality attenuates the positive association between internal audit and accounting information comparability” (pp.37) while the coefficient of the interaction term InterCon× IAQ is positive (0.023), and the authors conclude that “internal control quality fortifies the positive correlation between internal audit and accounting information comparability” (pp.40). That is, the interpretations are conflicting themselves. I would suggest authors have professional editing services to address the writing issue.

References:

Endrawes, M., Feng, Z., Lu, M., & Shan, Y. (2020). Audit committee characteristics and financial statement comparability. Accounting & Finance, 60(3), 2361-2395.

Guo, H., & Guo, H. (2021). Internal control and the comparability of accounting information - based on the mediating effect test of agency costs and information transparency. Friends of Accounting, 8, 79-86. (in Chinese)

Li, J., Xia, T., & Wu, D. (2022). Internal control quality, related party transactions and accounting information comparability. Procedia Computer Science, 199, 1252-1259.

Reviewer #3: I find it difficult to understand several sentences in the text. Therefore, I recommend proofreading and expanding them. This will ensure the content is clear, comprehensive, and easier to read and understand. This improved clarity will help convey the intended message more effectively.

6. PLOS authors have the option to publish the peer review history of their article (what does this mean?). If published, this will include your full peer review and any attached files.

Reviewer #1: No

Reviewer #2: No

Reviewer #3: No

---

## [Author Response · Author response to Decision Letter 0]

16 Jun 2024

Dear Reviewers,

Thank you for your helpful suggestions. Based on your constructive comments, we re-understand, re-think and strengthen the topic selection, writing logic and core issues of this article. Following your suggestions, we have further revised each part of the paper carefully, including the abstract, introduction, research design, research conclusions and managerial implication sections as follows.

---

## [Decision Letter · Decision Letter 1]

10 Sep 2024

Internal Audit Quality and Accounting Information Comparability: Evidence from China

PONE-D-24-12628R1

Dear Dr. Wang,

We’re pleased to inform you that your manuscript has been judged scientifically suitable for publication and will be formally accepted for publication once it meets all outstanding technical requirements.

Kind regards,

Muhammad Kaleem Khan

Academic Editor

PLOS ONE

Additional Editor Comments (optional):

Reviewers' comments:

Reviewer's Responses to Questions

**Comments to the Author**

1. If the authors have adequately addressed your comments raised in a previous round of review and you feel that this manuscript is now acceptable for publication, you may indicate that here to bypass the “Comments to the Author” section, enter your conflict of interest statement in the “Confidential to Editor” section, and submit your "Accept" recommendation.

Reviewer #1: All comments have been addressed

Reviewer #2: (No Response)

2. Is the manuscript technically sound, and do the data support the conclusions?

Reviewer #1: Yes

Reviewer #2: (No Response)

3. Has the statistical analysis been performed appropriately and rigorously? 

Reviewer #1: Yes

Reviewer #2: (No Response)

4. Have the authors made all data underlying the findings in their manuscript fully available?

Reviewer #1: Yes

Reviewer #2: (No Response)

5. Is the manuscript presented in an intelligible fashion and written in standard English?

Reviewer #1: Yes

Reviewer #2: (No Response)

6. Review Comments to the Author

Reviewer #1: The authors of the article "Internal Audit Quality and Accounting Information Comparability: Evidence from China" have seriously edited, completed, and supplemented according to the reviewers' comments. The article can be considered for publication.

Reviewer #2: (No Response)

7. PLOS authors have the option to publish the peer review history of their article (what does this mean?). If published, this will include your full peer review and any attached files.

Reviewer #1: No

Reviewer #2: No

---

## [Editor Report · Acceptance letter]

10 Oct 2024

PONE-D-24-12628R1 

PLOS ONE

Dear Dr. Wang, 

I'm pleased to inform you that your manuscript has been deemed suitable for publication in PLOS ONE. Congratulations! Your manuscript is now being handed over to our production team.

Kind regards, 

on behalf of

Dr. Muhammad Kaleem Khan 

Academic Editor

PLOS ONE